# Single-Side Superhydrophobicity in Si_3_N_4_-Doped and SiO_2_-Treated Polypropylene Nonwoven Webs with Antibacterial Activity

**DOI:** 10.3390/polym14142952

**Published:** 2022-07-21

**Authors:** Ming-Chao Han, Shun-Zhong Cai, Ji Wang, Hong-Wei He

**Affiliations:** 1Shandong Center for Engineered Nonwovens, Industrial Research Institute of Nonwovens & Technical Textiles, College of Textiles & Clothing, Qingdao University, Qingdao 266071, China; 19020636@qdu.edu.cn (M.-C.H.); 2021020789@qdu.edu.cn (S.-Z.C.); 2020020728@qdu.edu.cn (J.W.); 2State Key Laboratory of Bio-Fibers and Eco-Textiles, Qingdao University, Qingdao 266071, China

**Keywords:** PP meltblown nonwoven, silicon nitrile (Si_3_N_4_), filtration, antibacterial materials, superhydrophobicity

## Abstract

Meltblown (MB) nonwovens as air filter materials have played an important role in protecting people from microbe infection in the COVID-19 pandemic. As the pandemic enters the third year in this current global event, it becomes more and more beneficial to develop more functional MB nonwovens with special surface selectivity as well as antibacterial activities. In this article, an antibacterial polypropylene MB nonwoven doped with nano silicon nitride (Si_3_N_4_), one of ceramic materials, was developed. With the introduction of Si_3_N_4_, both the average diameter of the fibers and the pore diameter and porosity of the nonwovens can be tailored. Moreover, the nonwovens having a single-side moisture transportation, which would be more comfortable in use for respirators or masks, was designed by imparting a hydrophobicity gradient through the single-side superhydrophobic finishing of reactive organic/inorganic silicon coprecipitation in situ. After a nano/micro structural SiO_2_ precipitation on one side of the fabric surfaces, the contact angles were up to 161.7° from 141.0° originally. The nonwovens were evaluated on antibacterial activity, the result of which indicated that they had a high antibacterial activity when the dosage of Si_3_N_4_ was 0.6 wt%. The bacteriostatic rate against *E. coli* and *S. aureus* was up to over 96%. Due to the nontoxicity and excellent antibacterial activity of Si_3_N_4_, this MB nonwovens are promising as a high-efficiency air filter material, particularly during the pandemic.

## 1. Introduction

With the current COVID-19 pandemic still ongoing, there is a growing demand for functional materials, such as core filtration materials for respirators or masks [1,2]. At present, meltblown (MB) nonwovens made from polypropylene (PP) are the most widely used core layer for air filtration applications such as facial masks and air conditioning filter element [3]. The MB nonwovens are usually doped with materials and electrostatically charged after processing to optimize their filtration performance. Notably, quite a few studies have shown that respiratory diseases caused by bacteria and viruses are mostly transmitted by droplets or aerosols in the air [4,5,6,7]. Moreover, the long-term survival of bacteria and viruses on the surface of personal protective equipment (PPE) including face masks would cause potential risks of cross-contamination and postinfection to the users [8,9,10,11]. These put forward even higher requirements on the filtration performance of filter materials and at the same time, antibacterial functionalization has attracted increasing attention.

It is well-known that nonwoven surgical gown materials are typically finished with “three repellent and antistatic/directional water-transfer” to endow them with both performance and comfort [12,13]. The nonwovens can also be finished so as to make the inner side close to the skin possess a higher hydrophobicity, providing an effective means of achieving unidirectional moisture transfer [14,15]. The forming of micro/nano structural SiO_2_ could be ordinarily used as affordable fiber or fabric superhydrophobicity [16,17].

In general, the approaches to imparting antibacterial activity to nonwovens include surface finishing and blending modification of raw materials. The finishing process is commonly employed to introduce antibacterial agents into fabrics possessing a high strength, such as woven fabrics [18,19], spunlace nonwovens [20,21,22], needle-punched nonwovens [23] and spunbonded nonwovens [24,25]. In contrast with these fabrics, meltblown nonwovens are not suitable for the antibacterial finishing process due to their poor strength [26,27,28]. To prepare antibacterial meltblown nonwovens, it is more convenient and cost-effective to use blending modification, which applies antibacterial agents directly into a polymer raw material before web forming. The obtained antibacterial activity is also lasting. Limited by high-temperature MB processing, inorganic nanometals or metal oxides are generally used as antibacterial agents [29,30,31], and the silver compounds have especially been proved to be very effective. However, the potential hepatotoxicity of silver is still a concern, and its long-term effects on mammalian cells remain unclear [32,33]. The antibacterial mechanism of metal oxides is mainly photocatalytic, such as ZnO/PP antibacterial fibers [31], which hardly play an efficient role under less or no-light conditions. Ceramic materials have also been widely used in medical fields including dentistry, orthopedics and so on, due to its nontoxicity and lack of side effects. The antibacterial properties of ceramic materials have been gradually discovered recently. For example, the nanosized Si_3_N_4_ was coated or incorporated in polymer films as an antibacterial material, which showed good antibacterial activity [34,35], and its antibacterial mechanism is being gradually explored [36,37].

In this research, nano-Si_3_N_4_ ceramic materials were doped into PP by a twin-screw extruder to produce antibacterial masterbatches, which were transferred as raw material into antibacterial MB nonwovens during the meltblown process, which were further finished with superhydrophobicity on one side. The as-prepared nonwovens were evaluated on their filtration properties and antibacterial activities, with prospective applications in air filtration, including respirators and masks with a high-efficiency filtration performance and good antibacterial property.

## 2. Experimental

### 2.1. Materials

Polypropylene (melt flow rate, MFR: 1500 g/10 min, melting temperature: 165 °C) was kindly provided by Shandong Dawn SWT Technology Co., Ltd. (Yantai, China). Nano-Si_3_N_4_ (average particle size: 20 nm) was purchased from Shanghai Aladdin Biochemical Technology Co., Ltd. (Shanghai, China). Anhydrous ethanol (99.7%) and ammonia solutions (AR) were purchased from Sinopharm Chemical Reagent Co., Ltd. (Shanghai, China). Tetraethyl silicate (98%) was purchased from Macklin Biochemical Technology Co., Ltd. (Shanghai, China). The silane coupling agent (KH550) was purchased from Hangzhou Jacic Chemical Co., Ltd. (Hangzhou, China). Nutritional AGAR CM107 and nutritional broth CM106 were purchased from Beijing Luqiao Technology Co., Ltd. (Beijing, China). *Escherichia coli* ATCC8739 and *Staphylococcus aureus* ATCC6538 were obtained from Shanghai Luwei Technology Co., Ltd. (Shanghai, China).

### 2.2. Preparation of PP/Si_3_N_4_ Antibacterial Pellets for MB

To obtain PP antibacterial pellets as MB raw material with nano-Si_3_N_4_ dispersed fully and evenly, the blending of nano-Si_3_N_4_ particles with PP was carried out in a Benchtop Twin-Screw Extrusion Pelletizing Line (LTE16-40+, Labtech Engineering Co., Ltd., Samutprakarn, Thailand) with two feeders.

Firstly, PP and Si_3_N_4_ were added into the main feeder and one side of the extruder, respectively, and extruded, cooled and pelletized; the antibacterial masterbatch containing 20 wt% of Si_3_N_4_ was prepared, as shown in Figure 1. Secondly, the other PP/Si_3_N_4_ pellets as raw materials for MB nonwovens containing 0.6%, 0.8% and 1.0% of Si_3_N_4_ were prepared according to Table 1 and named SiN_0.6_, SiN_0.8_ and SiN_1.0_. The processing parameters of the twin-screw extruder are shown in Table 2, mainly including temperature sets in different zones. The screw rotation speed of the main feeder was 80 r/min and that of the side feeder was dynamically adjustable to be adaptable for the dosage.

### 2.3. Preparation of Antibacterial MB Nonwovens

The MB nonwovens were prepared on an MB lab line (SH-RBJ, Shanghai Sunhoo automation equipment Co., Ltd., Shanghai, China) and the MB process is illustrated in Figure 2. The raw material pellets were fed into the hopper, melted in the screw extruder and then extruded through the spinneret. At the same time, the air flow with a high temperature was blown out through both sides of the spinneret to draft the melt flow. The microfibers were uniformly collected on the receiving device to form web. The main online parameters of the melt-blown process are shown in Table 3.

### 2.4. Superhydrophobic Finishing on One Side of As-Prepared Nonwovens

The superhydrophobic finishing was carried out by in situ synthesis of silicon dioxide and precipitated on the surface of the nonwovens [38,39]. Firstly, solution A was prepared by adding 4.5 mL of tetraethyl silicate (TEOS), 1.0 mL of silane coupling agent (KH550) and 40.5 mL of anhydrous ethanol to a 250 mL beaker, and then MB nonwoven fabric cut into 2 cm × 2 cm size was floated on the surface of solution A and gently stirred with magnetic stirrers for 1 h. A solution B containing 9 mL of deionized water, 1.5 mL of ammonia water and 34.5 mL of anhydrous ethanol was dropped into the solution A slowly in 1 h and held for 3 h. Finally, the nonwovens were dried in an oven at 130 °C for 5 min.

### 2.5. Morphologies, Structures and Properties

The morphologies of the obtained nonwoven were observed by a scanning electron microscope (SEM, Phenom Pro, Eindhoven, The Netherlands). About 2.0 mm × 2.0 mm nonwoven sample bonded to a conductive tape was sprayed by gold for 60 s in an SBC-12 small ion sputtering instrument (Beijing, China). Then, it was observed in the SEM and under a 10 kV of electron accelerating voltage.

Nano Measurer 1.2.5 software was used to measure the fiber diameter and to calculate the average diameter of fibers in the as-produced nonwoven web and their diameter distributions.

The structures of raw materials and as-produced nonwovens were characterized by means of Fourier transform infrared spectroscopy (FTIR). The nano-Si_3_N_4_ was treated by the pressing disk method before tested and the as-produced nonwoven was tested in attenuated total reflection (ATR) mode.

The contact angle of the nonwoven material was measured by a contact angle analyzer (JY-PHb, Jinhe, China). Firstly, 5 μL of water was dropped on the surface of the nonwoven material, and then it was photographed. Finally, the contact angle was measured by the angle measurement method.

The thermal properties of antibacterial masterbatches or nonwovens were analyzed on a differential scanning calorimeter (DSC, Q2000, TA instruments, New Castle, DE, USA). The 5 mg sample was weighed and heated from 25 °C to 250 °C at a rate of 10 °C/min in a nitrogen atmosphere of 50 mL/min. The temperature was kept for 10 min and then cooled to 25 °C at the same rate. Moreover, the thermogravimetric analyzer (TGA) chart was also given from 25 °C to 750 °C at a rate of 10 °C/min in a nitrogen atmosphere of 50 mL/min.

### 2.6. Filtration Test

The pore size distribution of MB nonwovens was measured using the pore size meter TOPAS PSM-165 (Frankfurt, Germany). All samples were 25 g/m^2^ in basis weight, cut into disc-like samples with a diameter of 25 mm, put on a fixture with an internal cross-sectional area of 201 mm^2^ and then tested under an incrementally increasing airflow up to 70 L/min.

Air breathability was tested on the air permeability tester TEXTEST FX3300-IV (Schwerzenbach, Switzerland) under an air pressure of 200 Pa using a round sample having a surface area of 20 cm^2^.

The filtration performance of MB nonwoven was measured by the particle filter media test system TOPAS AFC-131 (Frankfurt, Germany). The sodium chloride aerosol particles were generated by the polydisperse aerosol generator, and its concentration was 1.0 mg/m^3^. The effective filtration area was 200 cm^2^ and the air flow rate was 5 m^3^/h.

### 2.7. Antibacterial Test

The antibacterial properties of SiN_0.6_, SiN_0.8_ and SiN_1.0_ nonwovens were evaluated by the oscillation method according to GB/T 20944.3-2008.

## 3. Results and Discussion

### 3.1. Fiber Morphologies, Structures and Properties of MB Nonwovens

As shown in Figure 3, the average fiber diameters of PP, SiN_0.6_, SiN_0.8_ and SiN_1.0_ nonwovens were 2.0 μm, 2.1 μm, 2.2 μm and 2.8 μm. Occasional coarse fibers were also observed in these nonwoven webs, possibly due to large particles or defect aggregates of Si_3_N_4_ causing a local blockage of the spinneret more or less, so that the melt flow was ejected unevenly. Overall, with the increase of Si_3_N_4_ content, the viscosity of the masterbatches was increased and MFR decreased (Appendix A), indicating a deteriorated fluidity of the PP melt and ultimately leading to thickened fiber diameters in the nonwoven webs.

### 3.2. Contact Angles (CAs) of MB Nonwovens

Owing to the uniform and well-distributed web of MB nonwovens, the so-called upper side and lower one of nonwovens had similar properties. The SiO_2_-treated side of nonwoven web in contact with the finishing solution was called the lower side and the other side called the upper side. The solution and process of hydrophobic finishing were shown in Figure 4a,b and the hydrophobic finishing occurred only on the lower side of nonwoven because of PP having a lower density than water and having a hydrophobicity with a CA of about 142°. Micro-nano structural SiO_2_ was formed and in situ precipitated on the surface of the nonwoven (Figure 4c), which afforded the nonwoven superhydrophobicity and the CA was increased significantly from 142.3° to 161.7° (Figure 3e and Figure 4g). The CA of the upper side was little changed (Figure 4d,f). It was indicated that the micro-nano structural SiO_2_ growing on the fibers’ surface led to chemical and structure hydrophobicity [39].

### 3.3. Thermal Properties of MB Nonwovens

The as-prepared nonwovens were also characterized by means of a DSC-TGA, and as shown in Figure 5, it indicated the thermal performance curves of nonwoven materials with different Si_3_N_4_ contents. The melting points of the PP, SiN0.6, SiN0.8 and SiN1.0 nonwovens were 166.9 °C, 167.3 °C, 167.1 °C and 167.3 °C (Figure 5a), which did not change very much, and all completely melted at above 180 °C of processing temperature. The decomposition temperatures based on 5% of weight loss were reached (Figure 5d), which were 414.9 °C, 413.3 °C, 411.4 °C and 413.2 °C for the PP, SiN0.6, SiN0.8 and SiN1.0 nonwovens, respectively. In general, the addition of Si_3_N_4_ did not change the melting point and decomposition temperature of PP obviously. It could be completely melted at 230 °C and would not be decomposed, which showed that the processing conditions in Table 3 were suitable.

### 3.4. Structures Analysis (FT-IR) of MB Nonwovens

As shown in Figure 6, the characteristic peaks of the –CH_3_ stretching vibration of PP were at 2949 cm^−1^ and 2867 cm^−1^ and those of the –CH_2_– stretching vibration at 2917 cm^−1^ and 2839 cm^−1^. The peaks at 1455 cm^−1^ and 1376 cm^−1^ were the bending vibration of the C–H bonds. The peaks at 1035 cm^−1^ and 923 cm^−1^ were attributed to the Si–N vibration, and the peaks at 578 cm^−1^ and 441 cm^−1^ to the Si–O vibration. Due to the low content, these peaks of Si_3_N_4_ did not appear in the nonwovens, which indicated that the addition of Si_3_N_4_ did not affect the structures of PP.

### 3.5. Pore Size and Air Permeability

The pore size and permeability of as-prepared nonwovens was tested to investigate the effect of nano Si_3_N_4_ on them. As shown in Figure 7, the average pore sizes of PP, SiN_0.6_, SiN_0.8_ and SiN_1.0_ nonwovens were 15.9 μm, 16.2 μm, 16.9 μm and 17.5 μm, and the air permeability 535 mm/s, 537 mm/s, 540 mm/s and 544 mm/s. Pure PP nonwoven had the narrowest pore size distribution, while SiN1 nonwoven had the widest pore size distribution. With the increase of Si_3_N_4_ content, the average diameter of nonwoven fibers became coarse and the diameter distribution became wide, which increased the permeability.

### 3.6. Filtration Performance

The filtration performance of MB nonwovens is generally indicated by its quality factor (QF) calculated based on the following Equation (1).
(1)QF=−ln(1−η)/Δp
where *η* represents the filtration efficiency in 100%, and ∆*p* represents the pressure drop in Pa.

The filtration efficiency of PP, SiN_0.6_, SiN_0.8_ and SiN_1.0_ nonwovens was 96.3%, 96.1%, 95.7% and 95.2%, respectively, and the pressure drops were 22.2 Pa, 21.4 Pa, 21.1 Pa and 20.5 Pa, respectively (Figure 8a). Accordingly, the filtration quality factors were 0.149 Pa^−1^, 0.152 Pa^−1^, 0.149 Pa^−1^ and 0.148 Pa^−1^ (Figure 7b), respectively, which stated that the addition of Si_3_N_4_ decreased the filtration efficiency and pressure drop. Based on Figure 7 and Figure 8, it was concluded that the fibers of nonwovens became coarser with the increase of Si_3_N_4_ content, which led to the increase of material pore size, and finally resulted in the decrease of the filtration performance.

### 3.7. Antibacterial Properties

The antibacterial properties of nonwovens, SiN_0.6_, SiN_0.8_ and SiN_1_ were evaluated according to GB/T 20944.3-2008, and the antibacterial function of Si_3_N_4_ was explored. Its bacteriostatic rate was calculated following Equation (2):(2)$$Y=\frac{W_{t}-Q_{t}}{W_{t}}\times 100\%$$
where Y is the bacteriostatic rate of the samples, and W_t_ is the mean value of the living bacteria concentration in the beaker after 18 h of oscillation contact of three pairs (CUF/mL). Q_t_ is the mean value of the living bacteria concentration in the flask after 18 h of oscillation contact of three antibacterial samples (CUF/mL). The evaluation results were shown in Table 4.

SiN_0.6_, SiN_0.8_ and SiN_1.0_ nonwovens had bacteriostatic rates of 97.01%, 97.13% and 97.17%, respectively, against *E. coli*. The antibacterial rates of antibacterial nonwovens containing 0.6%, 0.8% and 1.0% Si_3_N_4_ against *Staphylococcus aureus* were 97.09%, 97.17% and 97.21%, respectively, which showed that PP nonwovens doped with 0.6% of nano Si_3_N_4_ had good antibacterial activity. There are three antibacterial mechanisms of Si_3_N_4_. Firstly, the electric charge carried on the surface inhibits the growth of bacteria. Secondly, Si_3_N_4_ can change the wettability of the material and destroy the environment for bacterial reproduction. At last, Si_3_N_4_ reacts with water in the environment, in which some products drive bacterial lysis through chemical reactions on the surface [40]. The entire reaction of chemisorption, oxidation and dissolution on the surface of Si_3_N_4_ is shown in Equations (3) and (4) below.
(3)$$Si_{3}N_{4}+6H_{2}O\to 3SiO_{2}+4NH_{3}\to 3SiO_{2}+2N_{2}+6H_{2}$$
(4)$$3SiO_{2}+6H_{2}O\to 3Si{\left(\mathrm{OH}\right)}_{4}$$

## 4. Conclusions

One novel type of antibacterial MB nonwovens was prepared by introducing nano-Si_3_N_4_ ceramic and single-sided superhydrophobicity. The side of nonwovens treated with superhydrophobicity by a sol-gel method of SiO_2_ could reach a contact angle (CA) of up to 161.7°, which helped improve the effect of single-side moisture transport. The introduction of the Si_3_N_4_ content did not change the properties or performance of the as-prepared MB nonwovens in terms of their morphologies, structures, thermal properties and pore size filtration effect, except it substantially enhanced the antibacterial performance. At only 0.6% of Si_3_N_4_ addition, the bacteriostatic rate against *Escherichia coli* and *Staphylococcus aureus* could be up to 97%, and with the increase of the content of Si_3_N_4_, the antibacterial activity would be higher. Due to its low toxicity, this antibacterial MB nonwoven could play an important role to protect people from bacteria or viruses, and it could be especially significant during the pandemic caused by COVID-19.

## Figures and Tables

**Figure 1 polymers-14-02952-f001:**
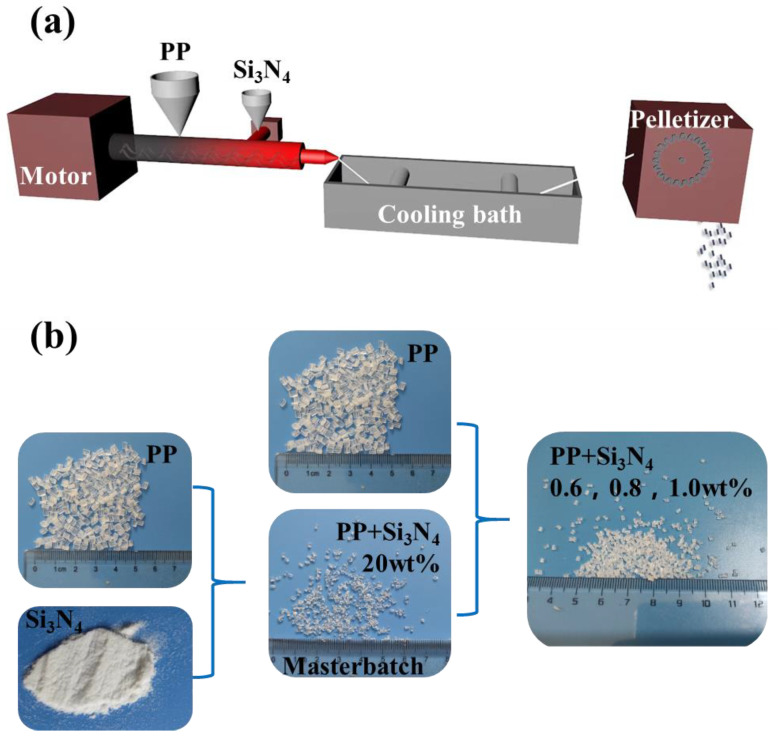
Preparation of PP/Si_3_N_4_ pellets: (**a**) twin-extruder process diagram, (**b**) PP/Si_3_N_4_ masterbatch and other pellets.

**Figure 2 polymers-14-02952-f002:**
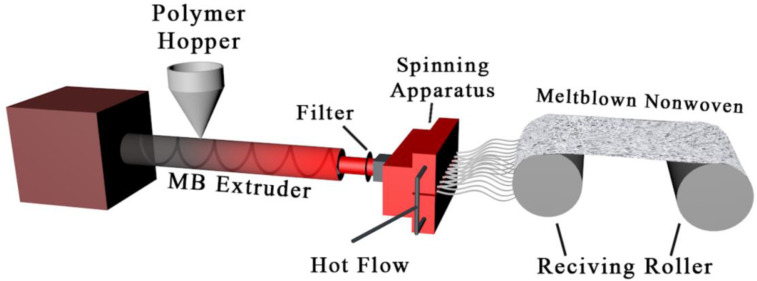
An illustration of the melt-blown process.

**Figure 3 polymers-14-02952-f003:**
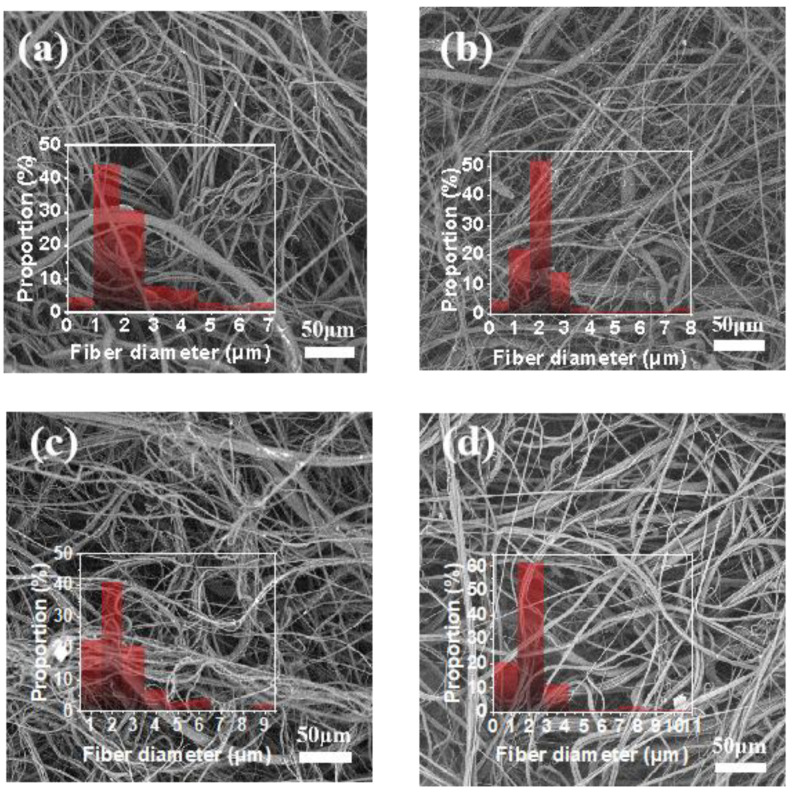
SEM images and fiber diameter distributions of the as-produced nonwovens with different Si_3_N_4_ content: (**a**) 0.0%, (**b**) 0.6%, (**c**) 0.8% and (**d**) 1.0%.

**Figure 4 polymers-14-02952-f004:**
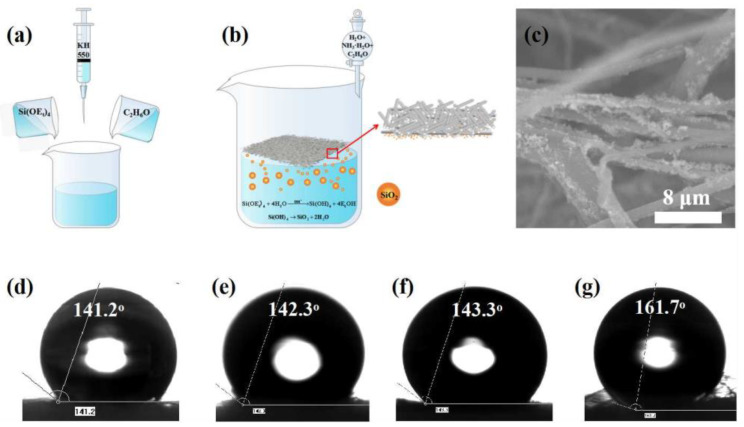
(**a**) Preparation of finishing solution, (**b**) hydrophobic finishing of SiO_2_ precipitation on lower side of nonwovens, (**c**) after the finishing, SEM photograph of lower side of nonwovens. The CAs of as-prepared MB nonwovens: (**d**) the upper side and (**e**) the lower side prior to finishing; correspondingly, (**f**) the upper side and (**g**) the lower side after the finishing.

**Figure 5 polymers-14-02952-f005:**
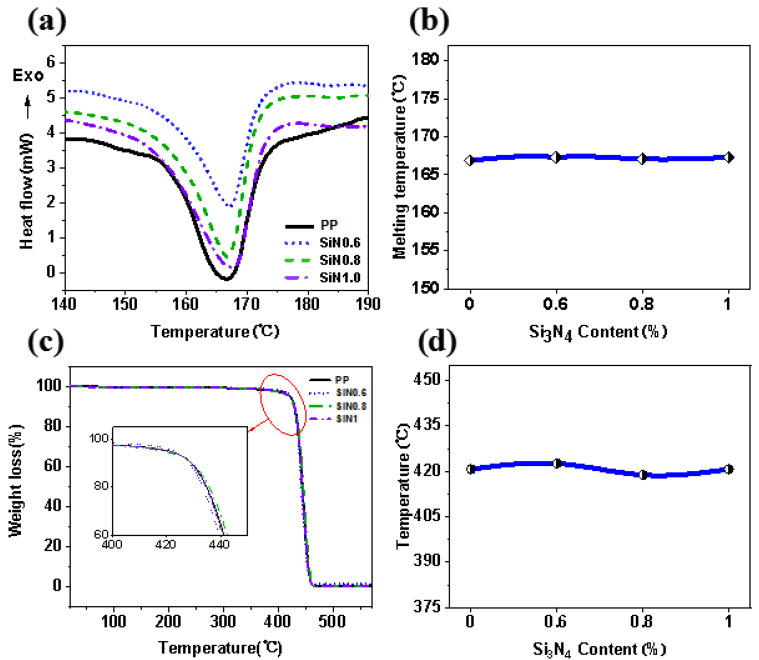
Thermal properties of PP with different Si_3_N_4_ contents: (**a**) heat flow curve (DSC), (**b**) melting temperature, (**c**) thermal decomposition curve and (**d**) decomposition temperature.

**Figure 6 polymers-14-02952-f006:**
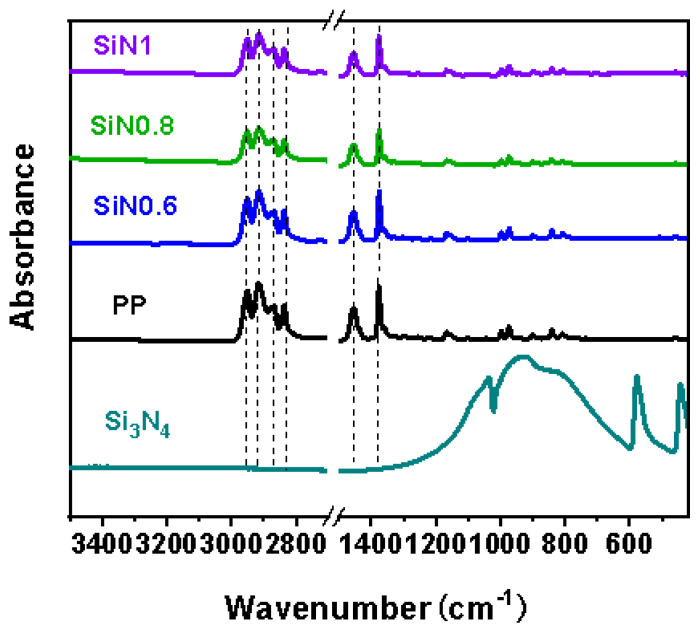
IR spectra of Si_3_N_4_ and PP with different Si_3_N_4_ contents.

**Figure 7 polymers-14-02952-f007:**
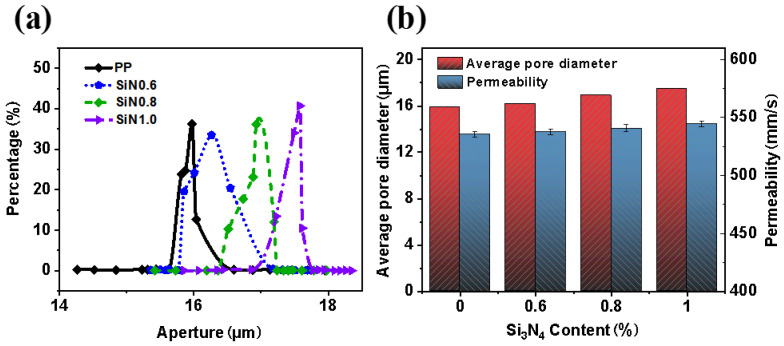
Pore size and permeability of nonwovens with different Si_3_N_4_ content: (**a**) pore size distribution and (**b**) average pore size and permeability.

**Figure 8 polymers-14-02952-f008:**
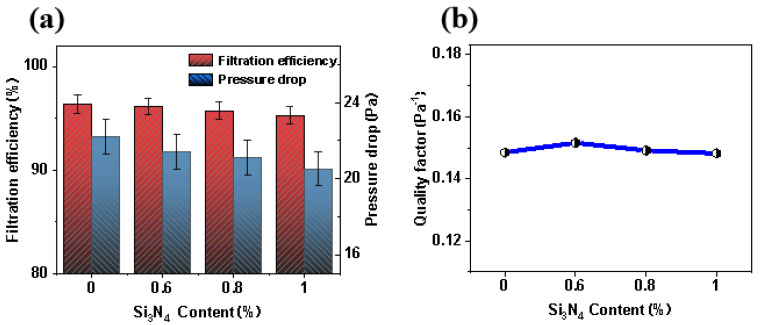
Filtration performance diagram of nonwoven materials with different Si_3_N_4_ content: (**a**) filtration efficiency and pressure drop; (**b**) quality factor.

**Table 1 polymers-14-02952-t001:** The recipe of raw materials for MB nonwovens.

Raw Materials	SiN_0.6_	SiN_0.8_	SiN_1.0_
PP, %	97	96	95
Antibacterial masterbatch (20% Si_3_N_4_)	3	4	5

**Table 2 polymers-14-02952-t002:** Processing parameters of twin-screw extruder.

Zone 1 and 2, °C	Zone 3 and 4, °C	Zone 5 and 6, °C	Zone 7 and 8, °C	Zone 9 and 10, °C
160	170	175	170	165

**Table 3 polymers-14-02952-t003:** The main online parameters of melt-blown process.

Zone 1, °C	Zone 2, °C	Zone 3, °C	Zone 4, °C	Spinneret, °C	Hot Air, °C	Air Pressure, MPa	Distance to Roller, cm
170	200	225	230	225	250	0.23	20

**Table 4 polymers-14-02952-t004:** Evaluation of antibacterial activity of as-prepared nonwovens.

Test Bacteria	(W_t_), CUF/mL	SiN_0_ (Q_t_), CUF/mL	SiN_0.6_ (Q_t_), CUF/mL	SiN_0.8_ (Q_t_), CUF/mL	SiN_1.0_ (Q_t_), CUF/mL	SiN_0_ (Y), %	SiN_0.6_ (Y), %	SiN_0.8_ (Y), %	SiN_1.0_ (Y), %
*Escherichia coli*	2.27 × 10^7^	2.21 × 10^7^	6.79 × 10^5^	6.51 × 10^5^	6.42 × 10^5^	2.64	97.01	97.13	97.17
*Staphylococcus aureus*	2.71 × 10^7^	2.62 × 10^7^	7.89 × 10^5^	7.68 × 10^5^	7.55 × 10^5^	3.32	97.09	97.17	97.21

## Data Availability

Not applicable.

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
