# Peer review of "Single-Side Superhydrophobicity in Si_3_N_4_-Doped and SiO_2_-Treated Polypropylene Nonwoven Webs with Antibacterial Activity"

_polymers, 2022, doi:10.3390/polym14142952_

Round 1

Reviewer 1 Report

Dear author,

Kindly go through all my comments

Author Response

  1. The title must be modified” The word good performance” doesn’t make sense

Authors reply:

Thank you very much.

Yes, we agree with you. These words would be deleted in the revised manuscript.

  1. What is the effect of ceramic materials on the flexibility or wearability of your product?

Authors reply:

Thank you very much.

The ceramic materials used in this article are nano powder solid, which is additive for the organic polymer (PP) matrix and doping content lower than 1wt%. The flexibility or wearability of nonwoven product was hardly affected.

  1. The durability test must be conducted say against washing

Authors reply:

Thanks a lot.

The application of as-prepared nonwovens in this research is filtration. On one hand, it will be employed as filter in personal protective equipment (PPE) including face masks, which is disposable product and generally the property of durability not considered. On the other hand, this kind of nonwovens can be also used as filter of air conditioner, which sometimes could be cleaned and installed again. In this case, this kind of addition antibacterial fabrics will show better durability than antibacterial finishing ones. The antibacterial activity will be higher as the antibacterial agents, nano particles will migrate to the surface of fiber [1-4].

[1] Gibbions N, Clarke N, Long D R. Migration of nanoparticles across a polymer-polymer interface: theory and simulation. Soft Matter, 2021,17:7294-7310.

[2] Bott J, Strmer A, Franz R. A model study into the migration potential of nanoparticles from plastics nanocomposites for food contact. Food Packaging & Shelf Life, 2014, 2(2):73-80.

[3] Störmer A, Bott J, Kemmer D, et al. Critical review of the migration potential of nanoparticles in food contact plastics. Trends in Food Science & Technology, 2017, 63:39-50.

[4] Bott J, Störmer A, Franz R. A model study into the migration potential of nanoparticles from plastics nanocomposites for food contact. Food Packaging & Shelf Life, 2014, 2(2):73-80.

  1. The effect of your treatment on comfort must be shown

Authors reply:

Thank you very much.

We agree that the comfort is important when the fabric applied in contact with the skin. As being used as core filters in PPE or air conditioners, the as-obtained nonwovens will not directly contact with the skin. And this property is not very important. In fact, the dosage of lower to 1wt% nano particles did not affect the intrinsic comfort of PP fibers.

  1. What mechanism have you employed so that the finishing agent will not penetrate to the other side of your sample

Authors reply:

Thank you very much.

As shown in the finishing process in the manuscript, thanks to the lower density of PP than that of water and intrinsic hydrophobic, the superhydrophobic finishing was occurred mainly on single side, as shown in below photos.

  1. I could prefer the full stop after citation like this (4). Not .(4)

Authors reply:

Thank you.

Yes, we agree with you. However, the citation format must be complied with the regulations of the publisher. We have noticed that both formats are adopted by different publishers, and MDPI (Polymers) adopt the latter.

Reviewer 2 Report

Comments:

Line 79: In the research the melting temperature of PP is important. This temperature should be given together with its MFI.

Line 92: It is unknown what means 16 mm for the twin-screw extrusion pelletizing line.

Lines 95-99: The description of the masterbatches preparation is not clear. Whether the blending of components was done two times?

Line 97: The Fig.S1 mentioned in the text is not attached. The figure is placed in supplementary file. This supplementary file is not extensive, so the figure can be attached to the main text.

Line 98: Which parameters are presented in Table 2? For the first extrusion or the second extrusion?  Why other formation parameters are not presented?

Line 103-106: Other formation parameters are missing. More details on melt-blown process are required.

Line 112. What does it mean as an example? Were there any other tests?

Lines 112; 140-142; 148: mL is not SI unit.

Line 115. Dimensions of samples are very small. Does it make sense to run such tests for such small samples? How many repetitions were carried out?

Lines 161-162: In Table 1 the masterbatches were signed. These labels should be used in the paper.

Line 166. The Fig.S2 from supplementary file can be attached to the main text. The results of the measurements of MFI of masterbatches should be attached to the main text too.

Fig.2. The histograms are hardly visible. The bars on the histograms should overlap (no gaps between).  

Lines 193-194; 216-217; 232-233; 254-255: Labelling was unnecessary?

Fig.4 The Fig.4c is strange. Why one curve is placed outside the coordinates?

Author Response

Line 79: In the research the melting temperature of PP is important. This temperature should be given together with its MFI.

Authors reply:

Thank you very much.

The melting temperature of PP is 165 °C, which has been added and marked in red in the revised manuscript.

Line 92: It is unknown what means 16 mm for the twin-screw extrusion pelletizing line.

Authors reply:

Thank you very much.

It is screw diameter. We have updated the information (model: LTE16-40+) of twin-extruder in the revised manuscript.

Lines 95-99: The description of the masterbatches preparation is not clear. Whether the blending of components was done two times?

Authors reply:

Thanks a lot.

Yes, the masterbatches were prepared by two steps. Firstly, PP and nano Si3N4 were mixed to prepare antibacterial masterbatch containing 20wt% of Si3N4. Secondly, the masterbatch containing 20wt% Si3N4 was mixed with pure PP to prepare masterbatches containing 0.6 wt%, 0.8 wt% and 1.0 wt% Si3N4, respectively. This process can make sure the nano particle of Si3N4 disperse into PP matrix more evenly. So, the process description was revised in the revised manuscript.

Line 97: The Fig.S1 mentioned in the text is not attached. The figure is placed in supplementary file. This supplementary file is not extensive, so the figure can be attached to the main text.

Authors reply:

Thank you very much.

This diagram has been modified and added in the revised manuscript and give more pictures to show the preparation process of masterbatches.

Line 98: Which parameters are presented in Table 2? For the first extrusion or the second extrusion?  Why other formation parameters are not presented?

Authors reply:

Thank you very much.

The parameters, temperature sets for the first and second extruding are the same. Different is the screw rotation speed of two feeders to adapt to the dosage of Si3N4, which was described more clearly and marked in red in the revised manuscript.

Line 103-106: Other formation parameters are missing. More details on melt-blown process are required.

Authors reply:

Thank you very much.

Meltblown (MB) process was described in detail and other online parameters had been added and marked in red in the revised manuscript. And more process parameters were attached in the Table 3.

Line 112. What does it mean as an example? Were there any other tests?

Authors reply:

Thank you very much.

The superhydrophobic finishing is a general method [38,39], and the expression has been revised and marked in red in the revised manuscript.

Lines 112; 140-142; 148: mL is not SI unit.

Authors reply:

Thank you very much.

Although the mL is not SI unit, it is more widely applied because liter (L, SI) is a considerably unit for lab-scale synthesis or test, just like length unit of mm, cm, μm and nm used instead of m (SI unit) in many articles.

Line 115. Dimensions of samples are very small. Does it make sense to run such tests for such small samples? How many repetitions were carried out?

Authors reply:

Thank you very much.

Several samples with similar size were superhydrophobic finishing in the same solution, and the results, contact angles were approximately similar as well. And the experiments were repeated 5 times, which was not abnormal.

Lines 161-162: In Table 1 the masterbatches were signed. These labels should be used in the paper.

Authors reply:

Thank you very much.

The expression has been revised and marked in the revised manuscript.

Line 166. The Fig.S2 from supplementary file can be attached to the main text. The results of the measurements of MFI of masterbatches should be attached to the main text too.

Authors reply:

Thank you very much.

Due to the low dosage of Si3N4, the change of MFI was not obvious and did not result in other novel properties, which may not be of significance. Under overall arrangement and layout of the article, the Figure is left in the supplementary file.

Fig.2. The histograms are hardly visible. The bars on the histograms should overlap (no gaps between).  

Authors reply:

Thank you very much.

Figure 2 has been modified in the revised manuscript.

Lines 193-194; 216-217; 232-233; 254-255: Labelling was unnecessary?

Authors reply:

Thank you very much.

The expressions have been revised and marked in the revised manuscript.

Fig.4 The Fig.4c is strange. Why one curve is placed outside the coordinates?

Authors reply:

Thanks a lot.

This is a combination diagram and the insert is amplification of assigned part of original diagram. It has been modified in the revised manuscript to clear up possible misunderstanding.

Round 2

Reviewer 1 Report

Well said

Author Response

Thank you very much.

Reviewer 2 Report

Comments

1/ The Fig.1 suggests that the extruded thread of masterbatch is introduced before cutting into a cooling bath. This fact should be described in the text.

2/ Usually during fibres or nonwoven formation the masterbatch is added to the pure polymer. It is unclear if the melt-blown nonwoven was formed from the masterbatch without adding the pure polymer granulate.  If yes using the term “masterbatch” for the used blend is confusing.

Author Response

Authors reply:

Thanks a lot.

The Fig.1 was modified in the revised manuscript, especially Fig.1a, the core parts of extruding line was labeled, such as, motor, cooling batch and pelletizer. The description was updated as well and marked in red in the revised manuscript.

2/ Usually during fibres or nonwoven formation the masterbatch is added to the pure polymer. It is unclear if the melt-blown nonwoven was formed from the masterbatch without adding the pure polymer granulate.  If yes using the term “masterbatch” for the used blend is confusing.

Authors reply:

Thank you very much.

The authors do agree with you. The term of masterbatch in this article cause easily confusing. As raw materials of other plastic or synthetic polymeric process, masterbatches mean polymeric pellets containing functional additives, generally with high content, and named after functional masterbatches, for instance, flame retardancy masterbatch, antibacterial masterbatch, and color masterbatch, etc, which are also applied in nonwoven fields.

For this work, the masterbatch should be mainly denoted as PP/Si3N4(20wt%). To describe more clearly, other kinds of PP/Si3N4(low content) materials were afforded as raw materials for nonwovens, which would be just called PP/Si3N4 pellets and named after SN series. Some descriptions were updated in the revised manuscript.

This manuscript is a resubmission of an earlier submission. The following is a list of the peer review reports and author responses from that submission.

Round 1

Reviewer 1 Report

Comments to the authors

Title: Single side superhydrophobicity in Si3N4 doped and SiO2 treated polypropylene nonwoven webs with antibacterial activity and good performances

ID: polymers-1733602

The article is written well and is under the scope of the journal. However, the following mandatory revisions must be done before considering publication in symmetry

  1. How is the exact functional finishing principle in abstract
  2. Remove the word antiviral in abstract where there is no use there
  3. Doe the functional finishing consider the change in behaviour of the pandemic
  4. Write up issues such as 2.1.1..
  5. It could be nice if you can draw the graph of abstract: subtitle 2.2
  6. Do not use ml instead write mL
  7. Lots of typographical errors like line subtitle 2.4 line 2
  8. What is the melting point of your chemicals as you made drying of nonwoven at 130 oC
  9. Durability against time or weather must be performed (The long-term stability of filtration efficiency must investigated at some length of day tracking test)
  10. Is any reason that you choose single sided coating for hydrophobicity
  11. Make short and precise introdcution

Reviewer 2 Report

The authors present a manuscript on the antibacterial activity of treated PP meltblown fibers with Si3N4 doped and SiO2. The authors mention the covid-19 pandemic in the abstract and in the conclusion  but they do not report any antiviral test for these materials. I think that the novelty of this manuscript in very limited and in my opinion do not deserve publication in Polymers considering both the limited novelty and the research approach proposed – I do not see evident polymer science in the paper. I think that this paper should  be submitted to other MDPI Journal such as Fibers or Materials.

-Concerning the manuscript, it seems a remake of a similar paper recently published from these authors https://doi.org/10.1007/s12221-022-4786-8

- Considering the kind of treatment it is evident that thermal properties (DSC) of PP are not affected, so these experiments are not useful in the main text.

- Are the authors sure that Si3N4 and SiO2 are not released from the fibers? Considering the kind of application, this is a crucial aspect and dedicated experiment should be designed and conducted.

- Antibacterial experiments should be conducted also on a control sample.

-the antibacterial protocol is based on 18h experiments; considering the kind of application proposed (respirators and masks), the antibacterical activity should be tested at least on the scale of minutes